# DEEP Q-LEARNING WITH LOW SWITCHING COST

## ABSTRACT

We initiate the study on deep reinforcement learning problems that require low switching cost, i.e., a small number of policy switches during training. Such a requirement is ubiquitous in many applications, such as medical domains, recommendation systems, education, robotics, dialogue agents, etc, where the deployed policy that actually interacts with the environment cannot change frequently. Our paper investigates different policy switching criteria based on deep Q-networks and further proposes an adaptive approach based on the feature distance between the deployed Q-network and the underlying learning Q-network. Through extensive experiments on a medical treatment environment and a collection of the Atari games, we find our feature-switching criterion substantially decreases the switching cost while maintains a similar sample efficiency to the case without the low-switching-cost constraint. We also complement this empirical finding with a theoretical justification from a representation learning perspective.

## 1 INTRODUCTION

Reinforcement learning (RL) is often used for modeling real-world sequential decision-making problems such as medical domains, personalized recommendations, hardware placements, database optimization, etc. For these applications, oftentimes it is desirable to restrict the agent from adjusting its policy frequently. In medical domains, changing a policy requires a thorough approval process by experts. For large-scale software and hardware systems, changing a policy requires to redeploy the whole environment. Formally, we would like our RL algorithm admits a *low switching cost*. In this setting, it is required that the deployed policy that interacts with the environment cannot change many times.

In some real-world RL applications such as robotics, education, and dialogue system, changing the deployed policy frequently may cause high costs and risks. Gu et al. (2017) trained robotic manipulation by decoupling the training and experience collecting threads; Mandel et al. (2014) applied RL to educational games by taking a data-driven methodology for comparing and validating policies offline, and run the strongest policy online; Jaques et al. (2019) developed an off-policy batch RL algorithms for dialog system, which can effectively learn in an offline fashion, without using different policies to interact with the environment. All of these work avoid changing the deployed policy frequently as they try to train the policy offline effectively or validate the policy to determine whether to deploy it online.

For RL problem with a low switching cost constraint, the central question is *how to design a criterion to decide when to change the deployed policy*. Ideally, we would like this criterion to have the following four properties:

1. **Low switching cost:** This is the purpose of this criterion. An algorithm equipped with this policy switching criterion should have low switching cost.

2. **High Reward:** Since the deployed policy determines the collected samples and the agent uses fewer deployed policies, the collected data may not be informative enough to learn the optimal policy with high reward. We need this criterion to deploy policies that can collect informative samples.

3. **Sample Efficiency:** Since the agent only uses a few deployed policies, there may be more redundant samples, which will not be collected if the agent switches the policy frequently. We would like algorithms equipped with a criterion with similar sample efficiency as the case without the low switching cost constraint.

4. **Generality:** We would like this criterion to be effective not only on a specific task but also broadly effective on a wide range of tasks.

In this paper, we take a step toward this important problem. We focus on designing a principled policy switching criterion for deep Q-networks (DQN) learning algorithms, which have been widely used in applications. For example, Ahn & Park (2020) apply DQN to control balancing between different HVAC systems, Ao et al. (2019) propose a thermal process control method based on DQN, and Chen et al. (2018) try to apply it to online recommendation. Notably these applications all require low switching cost.

Our paper conducts a systematic study on DQN with low switching cost. Our contributions are summarized below.

**Our Contributions**

- We conduct the first systematic empirical study on benchmark environments that require modern reinforcement learning algorithms. We test two naive policy switching criteria: 1) switching the policy after a fixed number of steps and 2) switching the policy after an increasing step with a fixed rate. We find that neither criterion is a generic solution because sometimes they either cannot find the best policy or significantly decrease the sample efficiency.

- Inspired by representation learning theory, we propose a new feature-based switching criterion that uses the feature distance between the deployed Q-network and the underlying learning Q-network. Through extensive experiments, we find our proposed criterion is a generic solution – it substantially decreases the switching cost while maintains a similar performance to the case without the low-switching-cost constraint.

- Along the way, we also derive a deterministic Rainbow DQN (Hessel et al., 2018), which may be of independent interest.

**Organization** This paper is organized as follows. In Section 2, we review related work. In Section 3, we describe our problem setup and review necessary backgrounds. In Section 4, we describe deterministic Rainbow DQN with the low switching cost constraint. In Section 5, we introduce our feature-based policy switching criterion and its theoretical support. In Section 6, we conduct experiments to evaluate different criteria. We conclude in Section 7 and leave experiment details to appendix.

## 2 RELATED WORK

Low switching cost algorithms were first studied in the bandit setting (Auer et al., 2002; Cesa-Bianchi et al., 2013). Existing work on RL with low switching cost is mostly theoretical. To our knowledge, Bai et al. (2019) is the first work that studies this problem for the episodic finite-horizon tabular RL setting. Bai et al. (2019) gave a low-regret algorithm with an $O\left(H^3 S A \log\left(K\right)\right)$ local switching upper bound where $S$ is the number of stats, $A$ is the number of actions, $H$ is the planning horizon and $K$ is the number of episodes the agent plays. The upper bound was improved in Zhang et al. (2020b;a).

The only empirical study on RL with switching cost is Matsushima et al. (2020), which proposed a concept of deployment efficiency and gave a model-based algorithm. During the training process, the algorithm fixes the number of deployments, trains a dynamics model ensemble, and updates the deployed policy alternately. After each deployment, the deployed policy collects transitions in the real environment to enhance the models, and then the models optimize the policy by providing imagined trajectories. In other words, they reduce the number of deployments by training on simulated environments. Our goal is different: we design a criterion to decide when to change the deployed policy, and this criterion could be employed by model-free algorithms.

There is a line of work on *offline* RL (also called Batch RL) methods, where the policy does not interact with the environment directly and only learns from a fixed dataset (Lange et al., 2012; Levine et al., 2020). Some methods Interpolate offline and online methods, i.e., semi-batch RL

algorithms (Singh et al., 1995; Lange et al., 2012), which update the policy many times on a large batch of transitions. However, the switching cost is not their focus.

## 3 PRELIMINARIES

### 3.1 MARKOV DECISION PROCESS

Throughout our paper, we consider the episodic Markov decision model $(\mathcal{S}, \mathcal{A}, H, P, \mathrm{r})$. In this model, $\mathcal{S}$ is the state space, $\mathcal{A}$ is the action space, $H \in \mathbb{Z}^+$ is the planning horizon. $P$ is the transition operator where $P(x'|x, a)$ denotes the transition probability of taking action $a$ from state $x$ to state $x'$. $r : \mathcal{S} \times \mathcal{A} \to \mathbb{R}$ is the reward function. A policy is a mapping from a state to an action, $\pi : \mathcal{S} \to \mathcal{A}$. In this paper, we focus on deterministic policies as required by motivating applications.

The dynamics of the episodic MDP can be view as the interaction of an agent with the environment periodically. We let $K$ be the total number of episodes the agent plays. At the beginning of an episode $k \in [K]$, the agent chooses a policy $\pi_k$. The initial $x_1^k \in \mathcal{S}$ is sampled from a distribution, and the agent is then in step 1. At each step $h \in [H]$ in this episode, based on the current state $x_h^k \in \mathcal{S}$ the agent chooses the action $a_h^k = \pi_k(x_h^k)$. The environment will give the reward for the step $r(x_h^k, a_h^k)$ and move the agent to the next state $x_{h+1} \sim P(\cdot|x_h^k, a_h^k)$. The episode automatically ends when the agent reaches the step $H + 1$.

$Q$-function is used to evaluate the long-term value for the action $a$ and subsequent decisions. The $Q$-function of a policy $\pi$ at time step $h$ is defined as follows:

$$Q_h^\pi(x, a) := r_h(x, a) + \mathbb{E}\left[\sum_{i=h+1}^{H} r\left(x_i, \pi\left(x_i\right)\right) \middle| x_h = x, a_h = a\right] \tag{1}$$

The goal of this agent is to find a policy $\pi^*$ which maximizes the expected reward, $\pi^* = \arg\max_\pi \mathbb{E}_\pi\left[\sum_{h=1}^{H} r_h\right]$. Ideally, we want to use a few episodes $(K)$ as possible to learn $\pi^*$.

### 3.2 SWITCHING COST

The concept of switching cost is used to quantify the adaptability of RL algorithms. The switching cost is defined as the number of policy changes of deployed policies in the running of the algorithm in $K$ episodes, namely:

$$N_{\text{switch}} := \sum_{k=1}^{K-1} \mathbb{I}\{\pi_k \neq \pi_{k+1}\} \tag{2}$$

The goal of this paper is to equip algorithm with a criterion that learns $\pi^*$ using a few episodes while at the same time has small $N_{\text{switch}}$.

### 3.3 DEEP $Q$-LEARNING

If we can estimate $Q$-function for each state and action pair well, there is no difficulty in finding $\pi^*$. For example, we can always select $a^* = \arg\max_a Q(x, a)$. However, it is not easy to learn Q value estimates for each state and action pair, especially when the state or the action space is large.

In deep $Q$-learning (DQN), Mnih et al. (2015) combine deep networks and reinforcement learning successfully by using a deep neural network to approximate $Q(x, a)$. Given the current state $x_h$, the agent selects a action $a_h$ greedily based on the $Q(x_h, a)$, then the state move to $x_{h+1}$ and a reward $r_{t+1}$ is obtained. The transition $(x_t, a_t, r_{t+1}, x_{t+1})$ is saved to the replay memory buffer. At each time, a batch of transitions is sampled from this buffer, and the parameters of neural networks are optimized by using stochastic gradient descent to minimize the loss

$$\left(r_{h+1} + \gamma_{h+1} \max_{a'} q_{\bar{\theta}}(x_{h+1}, a') - q_\theta(x_h, a_h)\right)^2 \tag{3}$$

where $\gamma_{h+1}$ is the chosen discount of time step $h + 1$, $\theta$ is the parameters of the online network, and $\bar{\theta}$ represents the parameters of the target network. The gradient of the loss is back-propagated

only to update $\theta$, while $\bar{\theta}$ is not optimized directly. DQN has been successful as it led to super-human performance on several Atari games. Nevertheless, there are also several limitations of this algorithm and many extensions have been proposed. Rainbow (Hessel et al., 2018) combines six of these extensions and obtains an excellent and stable performance on many Atari games. In the Appendix A, we review these tricks. In this paper, we focus on Rainbow DQN with low switching cost.

### 3.4 COUNT-BASED EXPLORATION

In many real-world scenarios that require low switching cost, it is also required to use deterministic policies, especially in applications mentioned above. Exploring strategies like $\epsilon$-greedy and noisy net make the policy stochastic, which we cannot use here. Count-based exploration algorithms are known to perform near-optimally when used in reinforcement learning for solving tabular MDPs. Tang et al. (2017) applied these algorithms into high-dimensional state spaces successfully. They discretized the state space with a hash function $\phi : \mathcal{S} \to \mathbb{Z}$. Then an exploration bonus $r^+(x) = \frac{\beta}{\sqrt{n(\phi(x))}}$ is added to the reward function and the agent is trained with the modified reward $r + r^+$. Note that with the count-based exploration, the policy is deterministic.

## 4 DETERMINISTIC RAINBOW DQN WITH A POLICY SWITCHING CRITERION

In this section, we first introduce how to implement a deterministic Rainbow DQN with a policy switching criterion. This implementation combines six DQN tricks and the policy is always deterministic.

### 4.1 DETERMINISTIC RAINBOW DQN

We first discuss how to make Rainbow DQN always output deterministic policies. Recall that to explore more efficiently, Rainbow adopts the Noisy Net (Fortunato et al., 2018), which makes the policy stochastic. To obtain a deterministic policy and keep exploring, we remove Noisy Net and employ the Count-Based exploration described in the previous section. When the deployed policy interacts with the environment, it selects the action which maximizes the $Q$-function. After obtaining the reward by taking this action, an exploration bonus $r^+ = \frac{\beta}{\sqrt{n(\phi(x))}}$ is added into the reward.

### 4.2 DQN WITH A POLICY SWITCHING CRITERION

Besides, Rainbow updates the policies which interacts with the environment directly. These policies usually switch millions of times during the training process (since the parameters are updated millions of times). In our implementation, we add an online policy in addition to the deployed policy that interacts with the environment. The deployed policy collects the data for the experience replay buffer, while the online policy is frequently updated when training. The deployed policy is replaced with the online policy when they meet the criterion we will discuss soon. The algorithm of deterministic switching Rainbow is shown in Algorithm 1.

### 4.3 POLICY SWITCHING CRITERIA

Here we first introduce two straightforward policy switching criteria.

**Fixed Interval Switching** This is the simplest criterion, we switch the deployed policy with a fixed interval. Under the *FIX_n* criterion, we switch the deployed policy whenever the online policy is updated $n$ times where $n$ is a per-specified number. We will specify $n$ in our experiments.

**Adaptive Interval Switching** This criterion aims to switch the deployed policy frequently at the first and reduce the switching speed gradually. Under the *Adaptive_n2m* criterion, we increase the deployment interval from $n$ to $m$. The interval between the $i$-th deployment and the $(i + 1)$-th deployment is $\min((i + 1) \times n, m)$. We will specify $n$ and $m$ in our experiments.

---

**Algorithm 1** Deterministic Switching Rainbow

---

1: Initialize arameters $\theta_{\text{online}}, \theta_{\text{deployed}}, \theta_{\text{target}}$ for online policy, deployed policy and target policy, initialize an empty replay buffer $D$
2: Denote the state encoder in online policy and deployed policy as $f_{\text{online}}$ and $f_{\text{deployed}}$
3: Set the step to start training $H_{\text{start}}$, the step to end training $H_{\text{max}}$, and the interval to update the target policy $H_{\text{target}}$.
4: Set $n(\text{accumulated\_updates}) = 0, n(\text{deployment}) = 0$
5: **for** h = 1 to $H_{\text{max}}$ **do**
6:     Select $a_h = \arg\max_a Q_{\text{deployed}}(s_h, a)$
7:     Execute action $a_t$ and observe reward $r_h$ and state $x_{h+1}$
8:     Compute the hash codes through for $x_h$, $\phi(x_h) = sgn(Ag(x_h))$
9:     ▷ $A$ is a fixed matrix with i.i.d. entries drawn from a standard Gaussian distribution $\mathcal{N}(0,1)$ and $g$ is a flat function
10:     Update the hash table counts, $n(\phi(x_h)) = n(\phi(x_h)) + 1$
11:     Update the reward $r_h = r_h + \frac{\beta}{\sqrt{n(\phi(r_h))}}$
12:     Store $(x_h, a_h, r_h, x_{h+1})$ in $D$
13:     **if** $h > H_{\text{start}}$ **then**
14:         Sample a minibatch of transitions from $D$.
15:         Update $\theta_{\text{online}}$ by stochastic gradient descent on the sampled minibatch once.
16:         **if** $h \quad \% \quad H_{\text{target}} == 0$ **then**
17:             Update $\theta_{\text{target}} = \theta_{\text{online}}$
18:             Set $n(\text{accumulated\_updates}) = n(\text{accumulated\_updates}) + 1$
19:         **end if**
20:         **if** $\mathcal{J}(f_{\text{deployed}}, f_{\text{online}}, D, n(\text{accumulated\_updates}), n(\text{deployment})) = true$ **then**
21:             Update $\theta_{\text{deployed}} = \theta_{\text{online}}$
22:             Update $n(\text{accumulated\_updates}) = 0$
23:             Update $n(\text{deployment}) = n(\text{deployment}) + 1$
24:         **end if**
25:     **end if**
26: **end for**

---

**Algorithm 2** Switching Criteria ($\mathcal{J}$ in Algorithm 1)

---

▷ Fixed interval switching *FIX_n*
**input** $n(\text{accumulated\_updates})$
**output** $bool(n(\text{accumulated\_updates}) \geq n)$

▷ Adaptive interval switching *Adaptive_n2m*
**input** $n(\text{accumulated\_updates}), n(\text{deployment})$
**output** $bool(n(\text{accumulated\_updates}) \geq \min(n(\text{deployment}) + 1) \times n, m)$

▷ Feature-based switching *F_a*
**input** $f_{\text{deployed}}, f_{\text{online}}, D$
Sample minibatch $\mathbb{B}$ of transitions $(x_h)$ from $D$ with probability $p_h$
Compute the the similarity $sim(x_h) = \frac{f_{\text{deployed}}(x_h) \cdot f_{\text{online}}(x_h)}{||f_{\text{deployed}}(x_h)|| \times ||f_{\text{online}}(x_h)||}$
Compute the average similarity in sampled batch $sim(\mathbb{B}) = \frac{\sum_{x \in \mathbb{B}} sim(x)}{||\mathbb{B}||}$
**output** $bool(sim(\mathbb{B}) \leq a)$

---

These two criteria and our proposed new criterion are summarized in Algorithm 2. Unfortunately, as will be shown in our experiments, these two criteria do not perform well. This requires us to design new principled criterion.

## 5    FEATURE-BASED SWITCHING CRITERION

In the section, we describe our new policy switching criterion based on feature learning. We first describe this criterion, and then we provide some theoretical justification from a representation learning point of view.

**Feature-based Switching Criterion**    We adopt the view the DQN learns to extract informative features of the states of environments. Our proposed criterion tries to switch the deployed policy according to the extracted feature. When deciding whether to switch the deployed policy or not, we first sample a batch of states $\mathbb{B}$ from the experience replay buffer, and then extract the feature of all states by both the deployed deep Q-network and online deep $Q$-network.

For a state $x$, the extracted feature are denoted as $f_{\mathrm{deployed}}(x)$ and $f_{\mathrm{online}}(x)$, respectively. The similarity score between $f_{\mathrm{deployed}}$ and $f_{\mathrm{online}}$ on state $x$ is defined as

$$sim(x) = \frac{\langle f_{\mathrm{deployed}}(x), f_{\mathrm{online}}(x) \rangle}{||f_{\mathrm{deployed}}(x)|| \times ||f_{\mathrm{online}}(x)||}$$

We then compute the averaged similarity score on the batch of states $\mathbb{B}$

$$sim(\mathbb{B}) = \frac{\sum_{x \in \mathbb{B}} sim(x)}{||\mathbb{B}||}$$

With a hyper-parameter $a \in [0, 1]$, the feature-based policy switching criterion is to change the deployed policy whenever $sim(\mathbb{B}) \leq a$.

**Theoretical Justification**    Our criterion is inspired by representation learning. To illustrate the idea, we consider the following setting. Suppose we want to learn $f(\cdot)$, a representation function that maps the input to a $k$-dimension vector. We assume we have input-output pairs $(x, y)$ with $y = \langle w, f^*(x) \rangle$ for some underlying representation function $f^*(\cdot)$ and a linear predictor $w \in \mathbb{R}^k$. For ease of presentation, let us assume we know $w$, and our goal is to learn the underlying representation which together with $w$ gives us $0$ prediction error.

Suppose we have data sets $\mathcal{D}_1$ and $\mathcal{D}_2$. We use $\mathcal{D}_1$ to train an estimator of $f^*$, denoted as $f^1$, and $\mathcal{D}_1 \cup \mathcal{D}_2$ to train another estimator of $f^*$, denoted as $f^{1+2}$. The training method is empirical risk minimization, i.e.,

$$f^1 \leftarrow \min_{f \in \mathcal{F}} \frac{1}{|\mathcal{D}_1|} \sum_{(x,y) \in \mathcal{D}_1} (y - \langle w, f(x) \rangle)^2 \text{ and } f^{1+2} \leftarrow \min_{f \in \mathcal{F}} \frac{1}{|\mathcal{D}_1 \cup \mathcal{D}_2|} \sum_{(x,y) \in \mathcal{D}_1 \cup \mathcal{D}_2} (y - \langle w, f(x) \rangle)^2$$

where $\mathcal{F}$ is some pre-specified representation function class.

The following theorem suggests if the similarity score between $f^1$ and $f^{1+2}$ is small, then $f^1$ is also far from the underlying representation $f^*$.

**Theorem 1.**  *Suppose $f^1$ and $f^{1+2}$ are trained via aforementioned scheme. There exist dataset $\mathcal{D}_1$, $\mathcal{D}_2$, function class $\mathcal{F}$ and $w$ such that if the similarity score between $f^1$ and $f^{1+2}$ on $\mathcal{D}_{1+2}$ is smaller than $\alpha$, then the prediction error of $f^1$ on $\mathcal{D}_{1+2}$ is $1 - \alpha$.*

The proof is deferred to Appendix B where we give explicitly constructions.

Theorem 1 suggests that in certain scenarios, if the learned representation has not converged (the similarity score is small), then it cannot be the optimal representation which in turn will hurt the prediction accuracy. Therefore, if we find the similarity score is small, we should change the deployed policy.

## 6    EXPERIMENTS

In this section, we conduct experiments to evaluate different policy switching criteria on DQN. We study several Atari game environments along and an environment for simulating sepsis treatment for ICU patients. We evaluate the efficiency among different switching criteria in these environments. Implementation details and hyper-parameters are listed in the Appendix A.

### 6.1 ENVIRONMENTS

**GYMIC**   GYMIC is an OpenAI gym environment for simulating sepsis treatment for ICU patients to an infection, where sepsis is caused by the body's response to an infection and could be life-threatening. GYMIC built an environment to simulate the MIMIC sepsis cohort, where MIMIC is an open patient EHR dataset from ICU patients. This environment generates a sparse reward, the reward is set to +15 if the patient recovered and -15 if the patient died. This environment has 46 clinical features and a $5 \times 5$ action space. For the GYMIC, we display the learning curve of 1.5 million steps of the environment, after which the reward converge. We choose this environment because it is simulating a real-world problem that requires low switching cost. For Atari games, all the experiments were training for the environment stepping 3.5 million times.

**Atari 2600**   Atari 2600 games are widely employed to evaluate the performance of DQN based agents. We also evaluate the efficiency among different switching criteria on several games, such as Pong, Road Runner, Beam Rider, etc.

### 6.2 RESULTS AND DISCUSSIONS

For all environments, we evaluate a feature-based criterion $F\_0.98$, three fixed interval criteria covering a vast range $FIX\_10^2, FIX\_10^3$ and $FIX\_10^4$, and an adaptive criterion increasing the deploying interval from 100 to 10,000. Besides the switching criteria we discussed above, we use "None" to indicate an experiment without the low-switching-cost constraint where deployed policy kept in sync with online policy all the time, notice that this experiment is equivalent to $FIX\_1$.

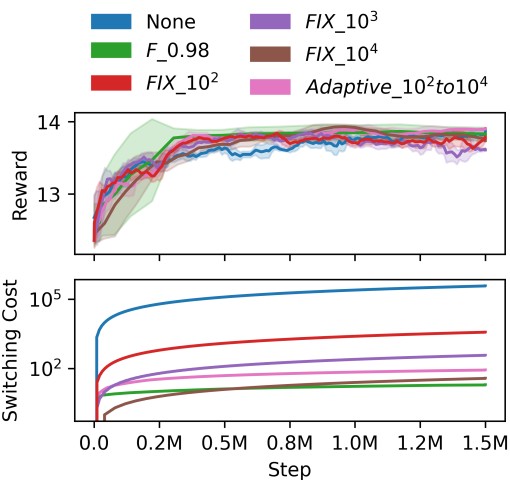

Figure 1: Results on GYMIC, "Step" means the number of steps of the environment. We show the learning curve of 1.5 million steps. The figure above is the learning cure of reward, while the figure below displays the switching cost. "None" means no low-switching-cost constraint, and the deployed policy always keeps sync with the online policy. Curves of reward are smoothed with a moving average over 5 points.

**GYMIC**   As shown in figure 1, none of the switching criteria affects the performance, but they can reduce the number of policy switches drastically, which indicates that reducing the switching cost in such a medical environment could be feasible. In particular, we find $FIX\_10^4$ and $F\_0.98$ are the two best criteria to reduce switching cost. Finally, this criterion keeps a good performance with the minimal switching cost.

**Atari 2600**   GYMIC may be too simple, and we should compare the performances among different criteria in some more difficult environments. We evaluate the performance of different criteria when playing the Atari games, which are image-based environments. In particular, the state space is much more complex. Figure 2 shows the results of 6 Atari games. In each subgraph, the upper curves are about the rewards of steps, while the curves blow is about switching cost.

First we observe that overall trend, higher switching cost leads to better performance. In general Rainbow DQN with no switching cost constrain often gives the best performance. Also, $FIX\_10^2$ enjoys better performance than $FIX\_10^3$ and $FIX\_10^4$. In some games such as Qbert and Riverraid, although $FIX\_10^4$ and $Adaptive$ lead to lower switching cost, they fail to learn how to play the games well. Therefore, they are not desired generic policy switching criterion.

Secondly, we observe that changing the policy with an adaptive interval may be better than a fixed interval. Focusing on the criteria $FIX\_10^4$ and $Adaptive\_10^2 to 10^4$, the adaptive criterion switches the online policy fast at first and decrease its switching speed gradually, and in the end, the adaptive criterion would have the same speed as the fixed one. Therefore, there is no significant difference

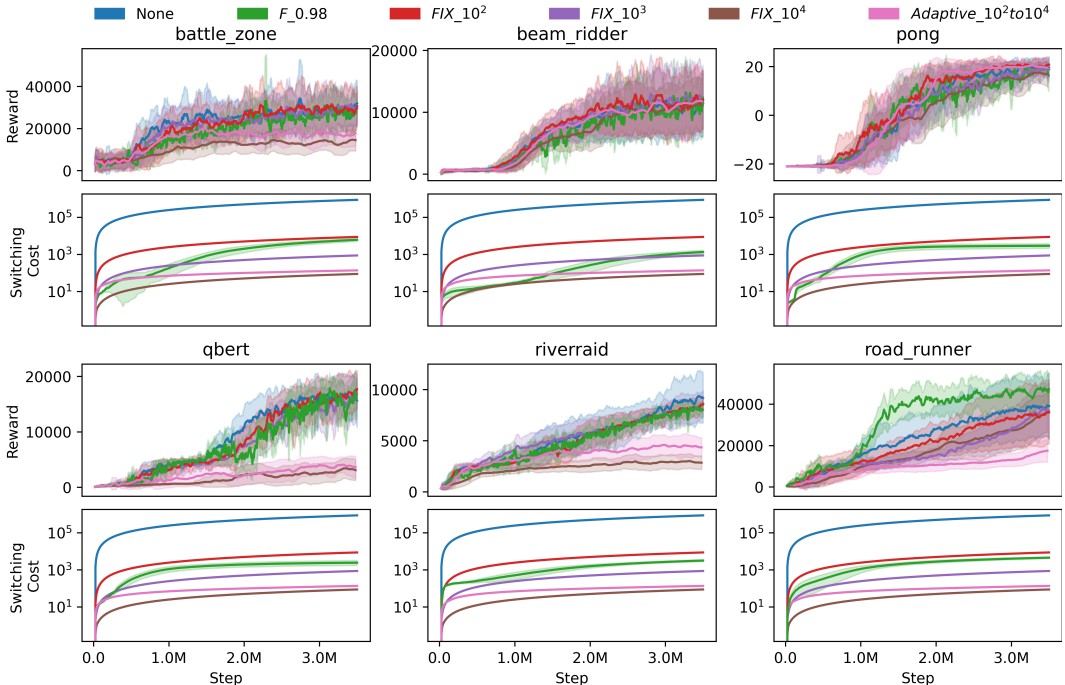

Figure 2: The results on the Atari games, we compare different switching criteria on six Atari games. "Step" means the number of steps of the environment. We constrain this 3.5 million steps for all environments. In each environment, we display the reward over the steps on the top and the switching cost in a $\log$ scale at the bottom. "None" means no switching criterion under which the deployed policy always keeps sync with the online policy. We evaluate a feature-based criterion, three fixed interval criteria covering a vast range, and an adaptive criteria increasing the deploying interval from 100 to 10,000. Curves of reward are smoothed with a moving average over 5 points.

between the total switching cost of these two criteria. However, we could observe that the adaptive criterion's performance is better than the fixed one when playing Beam Rider, Pong, Qbert, and they obtain similar performances in the rest games.

Lastly, we find our proposed feature-based criterion ($F\_0.98$) is the desired on that satisfy all four properties we discussed in Section 1. It significantly reduces the switching cost compared to "None", and is smaller than $FIX\_10^2$. While it incurs higher switching cost than $FIX\_10^3$, $FIX\_10^4$, and $Adaptive\_10^2 to 10^4$, *on all environments* feature-based criterion consistently perform as well as "None" in the sense that 1) it finds the optimal policy eventually, 2) it has the same sample efficiency as "None". On the other hand, other criteria sometime have significantly worse performance compared to "None", so none of them is a generic solution.

# 7    CONCLUSION

In this paper, we focus on the concept of switching cost and take a step toward designing a generic solution for reducing the switching cost while maintaining the performance. Inspired by representation learning theory, we proposed a new feature-based policy switching criterion for deep Q-learning methods. Through experiment on one medical simulation environment and six Atari games, we find our proposed criterion significantly reduces the switching cost and at the same time enjoys the same performance as the case where there is no switching cost constraint.

We believe our paper is just the first step on this important problem. One interesting question is how to design principled policy switching criteria for policy-based and model-based methods. Another direction is to give provable guarantees for these policy switching criteria that work for methods dealing with large state space in contrast to existing analyses are all about tabular RL (Bai et al., 2019; Zhang et al., 2020b;a).

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

# A DETAILS OF EXPERIMENTS

## A.1 DETAILED ALGORITHM

For completeness, we introduce the extensions of DQN and display our detailed algorithm.

**Double $Q$-learning**  Conventional Q-learning is affected by an overestimation bias, due to the maximization step in Equation 3, Double $Q$-learning (Hasselt et al., 2016) address this problem by decoupling. They use the loss

$$(r_{h+1} + \gamma_{h+1} q_{\bar{\theta}}(x_{h+1}, \arg\max_{a'} q_{\theta}(x_{h+1}, a')) - q_{\theta}(x_h, a_h))^2 \tag{4}$$

This change reduce harmful overestimations that were present for DQN, which leads to a improvement.

**Multi-step learning**  $Q$-learning accumulates a single reward, Sutton (2005) use a forward-view multi-step accumulated reward long ago, where a n-step accumulated reward is defined as

$$r_h^{(n)} := \sum_{k=0}^{n-1} \gamma_h^{(n)} r_{h+k+1} \tag{5}$$

and the final loss is

$$(r_h^{(n)} + \gamma_h^{(n)} \max_{a'} q_{\bar{\theta}}(x_{h+1}, a') - q_{\theta}(x_h, a_h))^2 \tag{6}$$

Multi-step targets often lead to faster learning (Sutton & Barto, 2005)

**Dueling networks**  Wang et al. (2016) splits the DQN network into two streams called value stream $V$ and advantage stream $A$, these two strems share the convolutional encoder $f$, and action value $Q(x, a)$ could be computed as:

$$Q(x, a) = V(f(x)) + A(f(x), a) - \frac{\sum_{a'} A(f(x), a')}{N_{\text{actions}}} \tag{7}$$

**Prioritized replay**  DQN samples uniformly from the replay buffer, to sample more frequently from those transitions from which the policy can learn more and quickly, Schaul et al. (2016) samples transitions with probability $p_h$ relative to the last encountered absolute TD error

$$p_h \propto |r_{h+1} + \gamma_{h+1} \max_{a'} q_{\bar{\theta}}(x_{h+1}, a') - q_{\theta}(x_h, a_h)|^\omega \tag{8}$$

where $\omega$ is a hyper-parameter, and new transitions always have the maximum priority when they enter the buffer to ensure a bias towards unseen transitions.

**Distributional RL**  Instead of approximating the expected return as DQN, Bellemare et al. (2017) proposed a method to approximate the distribution of returns on a a discrete support $z$, $z$ is a vector with $N_{atoms}$ atoms and is defined as

$$z_i = V_{\min} + (i-1)\frac{V_{\max} - V_{\min}}{N_{\text{atoms}} - 1}, i \in 1, 2, ..., N_{\text{atoms}} \tag{9}$$

where $V_{min}$ and $V_{max}$ is the minimal and maximal value in this support. The approximating distribution $d_h$ is defined on this support and the probability $p_\theta^i(x_h, a_h)$ on each atom $i$ as $d_h = (z, p_\theta(x_h, a_h))$ .The goal is to update the trainable parameters $\theta$ to match this distribution with the actual distribution of returns. To learn the probability $p_\theta^i$ for each $i$ with a variant of Bellman's equation, they minimize the Kullbeck-Leibler divergence $D_{KL}(\Phi_z d_h' || d_h)$ between the distribution $d_h$ and the target distribution $d_h' := (r_{h+1} + \gamma_{h+1} z, p_{\bar{\theta}}(x_{h+1}, \arg\max_{a'} q_{\bar{\theta}}(x_{h+1}, a')))$, where $\Phi_z$ is a L2-projection from the target distribution to the fixed support $z$ and $q_{\bar{\theta}}(x_{h+1}, a) = z^T p_{\bar{\theta}}(x_{h+1}, a)$

**Nosiy Net** To address the limitations of exploring using $\epsilon$-greedy policies, Fortunato et al. (2018) propose a noisy linear layer

$$y = b + Wx + (b_{noisy} \odot \epsilon^b + (W_{noisy} \odot \epsilon^w)x) \tag{10}$$

to replace the standard linear $y = b + Wx$, where $\epsilon^b$ and $\epsilon^w$ are random variables, $\odot$ denotes the element-wise product.

Rainbow combines all of these 6 extensions, To make the policy deterministic but still keep exploring during training, we remove the Noisy Net and adopt Count-Based exploration. The detailed algorithm is as follows.

---

**Algorithm 3** Detailed algorithm

---

1: ▷ Hyper-parameters for extensions
2: Distributional RL: Number of atoms $N_{\text{atoms}}$, min/max values $V_{\text{min}}/V_{\text{max}}$
3: Prioritization replay memory: exponent $\omega$, capacity $N$
4: Multi-step: number of steps $n$
5: ▷ Initialization
6: Initialize prioritization replay memory $D$, state encoder $f_{\text{online}}$, value stream $V_{\text{online}}$ and advantage stream $A_{\text{online}}$ for online policy $P_{\text{online}}$
7: Initialize deployed policy $P_{\text{deployed}}$ and target policy $P_{\text{target}}$ with parameters in $P_{\text{online}}$
8: ▷ Definition of Q-function
9: $z_i = V_{\text{min}} + (i-1)\frac{v_{\text{max}} - v_{\text{min}}}{N_{\text{atoms}} - 1}$
10: $\bar{A}^i_{\text{online}}(x) = \frac{1}{N_{\text{actions}}} \sum_a A^i_{\text{online}}(f_{\text{online}}(x), a)$
11: $p^i_{\text{online}}(x, a) = softmax(V^i_{\text{online}}(f_{\text{online}}(x)) + A^i_{\text{online}}(f_{\text{online}}(x), a) - \bar{A}^i_{\text{online}}(x))$
12: $Q_{\text{online}}(x, a) = \sum_i z_i p^i_{\text{online}}(x, a)$
13: Similarly, $Q_{\text{deployed}}(x, a) = \sum_i z_i p^i_{\text{deployed}}(x, a)$, $Q_{\text{target}}(x, a) = \sum_i z_i p^i_{\text{target}}(x, a)$
14:
15: ▷ Training process
16: set up environment
17: $n(\text{accumulated\_updates}) = 0, n(\text{deployment}) = 0$
18: **for** $h = 1$ to $H_{\text{max}}$ **do**
19:     **if** Termination **then**
20:         reset environment
21:     **end if**
22:     select $a_h = \arg\max_a Q_{\text{deployed}}(x_h, a)$
23:     Execute action $a_h$ in emulator and observe reward $r_h$ and state $x_{h+1}$
24:     compute the hash codes through for $x_h$, $\phi(x_h) = sgn(Ag(x_h))$
25:     ▷ $A$ is a matrix with i.i.d. entries drawn from a standard Gaussian distribution $\mathcal{N}(0, 1)$ and $g$ is a flat function
26:     update the hash table counts, $n(\phi(x_h)) = n(\phi(x_h)) + 1$
27:     update the reward $r_h = r_h + \frac{\beta}{\sqrt{n(\phi(x_h))}}$
28:     store $(x_h, a_h, r_h, x_{h+1})$ in $D$
29:     **if** $h > H_{\text{start}}$ **then**
30:         sample minibatch of transitions $(x_{h'}, a_{h'}, w_{h'}, r_{h'}, r_{h'+1} ..., r_{h'+n-1}, x_{h'+n})$ from $D$ with probability $p_{h'}$
31:         ▷ $w_{h'}$ is the importance-sampling weight for transitions at $h'$
32:         compute the multi-step reward $r^{(n)}_{h'} = \sum_{k=0}^{n-1} \gamma^n r_{h'+k}$
33:         $a^* = \arg\max_a Q_{\text{online}}(x_{h'+n}, a), m_i = 0, i \in 1, 2, ..., N_{\text{atoms}}$
34:         **for** $j = 1$ to $N_{\text{atoms}}$ **do**
35:             $\hat{\mathcal{T}}z_j = [r^{(n)}_{h'} + \gamma^n z_j].clip(V_{min}, V_{max})$
36:             $\Delta z = (V_{\text{max}} - V_{\text{min}})/(N_{\text{atoms}} - 1)$
37:             $b_j = (\hat{\mathcal{T}}z_j - V_{\text{min}})/\Delta z, l = \lfloor b_j \rfloor, u = \lceil b_j \rceil$
38:             $m_l = m_l + p^j_{\text{target}}(x_{h'+n}, a^*)(u - b_j), m_u = m_u + p^j_{\text{target}}(x_{h'+n}, a^*)(b_j - l)$
39:         **end for**
40:         $D_{KLh'} = -\sum_i m_i \log p^i_{\text{online}}(x_{h'}, a_{h'})$
41:         update $p_{h'}$ by $||D_{KLh'}||^\omega$
42:         update parameters in $P_{\text{online}}$ by a gradient descent step on $D_{KLh'} \times w_{h'}$

43:     $n(\text{accumulated\_updates}) = n(\text{accumulated\_updates}) + 1$
44:     **if** $h \quad \% \quad H_{\text{target}}$ **then**
45:         update $P_{\text{target}}$ by the parameters of $P_{\text{online}}$
46:     **end if**
47:     **if** $\mathcal{J}(f_d, f_o, D, n(\text{accumulated\_updates}), n(\text{deployment})) = true$ **then**
48:         update $P_{\text{deployed}}$ by the parameters of $P_{\text{online}}$
49:         $n(\text{accumulated\_updates}) = 0, n(\text{deployment}) = n(\text{deployment}) + 1$
50:     **end if**
51:     **end if**
52: **end for**

## A.2    HYPE-PARAMETERS

Table 1 lists the basic hyper-parameters of the algorithm, all of our experiments share these hyper-parameters except the experiments on GYMIC adopt the $H_{\text{target}}$ as $1K$. Most of these parameter are the same with raw Rainbow algorithm. For the count base exploration, the bonus is $\beta$ set to 0.01.

| Parameter | Value |
|---|---|
| $H_{\text{start}}$ | 20K |
| learning rate | 0.0000625 |
| $H_{\text{target}}$(Atari) | 8K |
| $H_{\text{target}}$(GYMIC) | 1K |
| Adam $\epsilon$ | $1.5 \times 10^{-4}$ |
| Prioritization type | proportional |
| Prioritization exponent $\omega$ | 0.5 |
| Prioritization importance sampling | $0.4 \rightarrow 1.0$ |
| Multi-step returns n | 3 |
| Distributional atoms $N_{\text{atoms}}$ | 51 |
| Distributional $V_{\text{min}}, V_{\text{max}}$ | [-10, 10] |
| Discount factor $\gamma$ | 0.99 |
| Memory capacity N | 1M |
| Replay period | 4 |
| Minibatch size | 32 |
| Reward clipping | [-1, 1] |
| Count-base bonus | 0.01 |
| Activation function $\beta$ | ReLu |

Table 1: The basic hyper-parameters, we used the Adam optimizer with learing rate $\alpha = 0.0000625$ and $\epsilon = 1.5 \times 10^{-4}$ , before training the online policy, we let the initialized random policy make 20K steps to collect some transitions and the capacity for replay buffer is 1M. During the training process, we sample 32 transitions from the replay buffer and update the online policy every four steps. The reward is clipped into [-1, 1] and Relu is adopted as the activation function. For replay prioritization we use the recommended proportional variant, with importance sampling from 0.4 to 1, the prioritization $\omega$ is set to 0.5. In addition, we employ $N_{\text{atoms}} = 51, V_{\text{min}} = -10, V_{\text{max}} = 10$ for distributional RL and $n = 3$ for multi-step returns. Finally the count-base bonus is set to 0.01

Table 2 lists the rest hyper-parameters for experiments on GYMIC. Since there are 46 clinical features in this environment, we stack 4 consecutive states to compose a 184-dimensional vector as the input for the state encoder $f_{\text{online}}(f_{\text{deployed}}$ or $f_{\text{target}})$. The state encoder is a 2-layer MLP with hidden size 128.

Table 3 shows the additional hyper-parameters for experiments on Atari games. The observations are grey-scaled and resized to $84 \times 84$ tensor, and 4 consecutive frames are concatenated as a single state, each action selected by the agent is repeated for 4 times. The state encoder is composed of 3 convolutional layers with 32, 64 and 64 channels, which use 8x8, 4x4, 3x3 filters and strides of 4, 2, 1 respectively.

| Parameter | Value |
|---|---|
| State Stacked | 4 |
| Number of layers for MLP | 2 |
| Hidden size | 128 |

Table 2: Extra hype-parameters for the experiments in GYMIC, we stack 4 consecutive states and adopt a 2-layer MLP with hidden size 128 to extract the feature of states.

| Parameter | Value |
|---|---|
| Gray scaling | True |
| Observation | (84, 84) |
| Frame Stacked | 4 |
| Action repetitions | 4 |
| Max frames per episode | 108k |
| Encoder channels | 32, 64, 64 |
| Encoder filter size | $8 \times 8, 4 \times 4, 3 \times 3$ |
| Encoder stride | 4, 2, 1 |

Table 3: Additional hyper-parameters for experiments in Atari games. Observations are grey-scaled and rescaled to $84 \times 84$ 4 consecutive frames are staked as the state and each action is acted four times. And we limit the max number of frames for an episode to 108K. The state encoder consists of 3 convolutional layers.

### A.3 REWARD AND SWITCHING COST

In the end, we list the value of the switching cost and reward of different criteria when the environment take 1.5 million steps and 3 million steps in Table 4.

## B PROOF FOR SECTION 5

*Proof of Theorem 1.* We let $w = (1, 1, \ldots, 1) \in \mathbb{R}^k$ be a $k$-dimensional all one vector. We let

$$\mathcal{F} = \{f : f(x) = (2\sigma(\langle v_1, x \rangle) - 1, 2\sigma(\langle v_2, x \rangle) - 1, \ldots, 2\sigma(\langle v_k, x \rangle) - 1)\} \subset \{\mathbb{R}^k \to \mathbb{R}^k\}$$

with $\sigma(\cdot)$ being the ReLU activation function[1] and $v_i \in \{e_i, -e_i\}$ where $e_i \in \mathbb{R}^k$ denotes the vector that only the $i$-th coordinate is 1 and others are 0. We assume $k$ is an even number and $\alpha k$ is an integer for simplicity. We let the underlying $f^*$ be the vector correspond to $(e_1, e_2, \ldots, e_k)$. We let $\mathcal{D}_1 = \{(e_1, 1), (e_2, 1), \ldots, (e_{(1-\alpha)k}, 1)\}$ and $\mathcal{D}_2 = \{(e_{(1-\alpha)k+1}, 1), \ldots, (e_k, 1)\}$. Because we use the ERM training scheme, it is clear that the training on $\mathcal{D}_1 \cup \mathcal{D}_2$ will recover $f^*$, i.e., $f^{1+2} = f^*$ because if it is not $f^*$ is better solution ($f^*$ has 0 error ) for the empirical risk. Now if the similarity score between $f^1$ and $f^{1+2}$ is smaller than $\alpha$, it means for $f^1$, its corresponding $\{v_{(1-\alpha)k+1}, \ldots, v_k\}$ are not correct. In this case, $f^1$'s prediction error is at least $1 - \alpha$ on $\mathcal{D}_1 \cup \mathcal{D}_2$, because it will predict 0 on all inputs of $\mathcal{D}_2$.

$\square$

---

[1]We define $\sigma(0) = 0.5$

| environment | criteria | None | FIX.$10^2$ | FIX.$10^3$ | FIX.$10^4$ | Adaptive | F.0.98 |
|---|---|---|---|---|---|---|---|
| | | | **1,500,000 steps** | | | | |
| battle_zone | Switching cost | 368750.0 | 3687.5 | 369.0 | 36.5 | 84.5 | 529.2 |
| | Reward | 26900.0 ± 9445.1 | 23070.0 ± 7691.9 | 17930.0 ± 4472.7 | 18870.0 ± 4417.4 | 14330.0 ± 4397.9 | 16970.0 ± 5766.2 |
| | Gap | N/A | -3830.0 | -8970.0 | -8030.0 | -12570.0 | -9930.0 |
| beam_rider | Switching cost | 368750.0 | 3687.5 | 369.0 | 36.5 | 84.5 | 119.6 |
| | Reward | 7655.9 ± 2021.2 | 7554.26 ± 2283.4 | 7493.76 ± 2376.4 | 4068.26 ± 1340.2 | 4078.56 ± 1819.8 | 6759.74 ± 2136.3 |
| | Gap | N/A | -101.6 | -162.1 | -3587.6 | -3577.3 | -896.2 |
| pong | Switching cost | 368750.0 | 3687.5 | 369.0 | 36.5 | 84.5 | 1681.4 |
| | Reward | −12.87 ± 8.1 | −2.5 ± 9.2 | −13.69 ± 4.0 | −12.97 ± 5.2 | −7.14 ± 10.2 | 0.08 ± 6.7 |
| | Gap | N/A | 10.4 | -0.8 | -0.1 | 5.7 | 12.9 |
| qbert | Switching cost | 368750.0 | 3687.5 | 369.0 | 36.5 | 84.5 | 1268.5 |
| | Reward | 6050.25 ± 2189.2 | 5073.75 ± 624.7 | 4359.75 ± 534.2 | 649.25 ± 245.3 | 2681.0 ± 1696.1 | 3947.5 ± 1438.0 |
| | Gap | N/A | -976.5 | -1690.5 | -5401.0 | -3369.2 | -2102.8 |
| riverraid | Switching cost | 368750.0 | 3687.5 | 369.0 | 36.5 | 84.5 | 899.5 |
| | Reward | 5453.2 ± 787.6 | 4602.8 ± 693.8 | 4177.0 ± 870.0 | 3152.6 ± 658.6 | 3267.6 ± 705.2 | 4182.2 ± 925.8 |
| | Gap | N/A | -850.4 | -1276.2 | -2300.6 | -2185.6 | -1271.0 |
| road_runner | Switching cost | 368750.0 | 3687.5 | 369.0 | 36.5 | 84.5 | 1309.0 |
| | Reward | 13266.0 ± 1513.4 | 26039.0 ± 4605.4 | 14947.0 ± 2000.3 | 30620.0 ± 6946.2 | 19513.0 ± 4751.0 | 28968.0 ± 4827.2 |
| | Gap | N/A | 12773.0 | 1681.0 | 17354.0 | 6247.0 | 15702.0 |
| | | | **3,000,000 steps** | | | | |
| battle_zone | Switching cost | 743750.0 | 7437.5 | 744.0 | 73.5 | 122.5 | 4240.3 |
| | Reward | 32250.0 ± 8119.6 | 23910.0 ± 6289.8 | 21290.0 ± 8191.8 | 17830.0 ± 4384.2 | 12530.0 ± 5096.0 | 26180.0 ± 7161.5 |
| | Gap | N/A | -8340.0 | -10960.0 | -14420.0 | -19720.0 | -6070.0 |
| beam_rider | Switching cost | 743750.0 | 7437.5 | 744.0 | 73.5 | 122.5 | 1233.4 |
| | Reward | 13751.2 ± 6530.9 | 11393.24 ± 4802.1 | 12236.62 ± 5669.9 | 5654.16 ± 2120.0 | 8857.94 ± 3173.9 | 12311.34 ± 6731.4 |
| | Gap | N/A | -2358.0 | -1514.6 | -8097.0 | -4893.3 | -1439.9 |
| pong | Switching cost | 743750.0 | 7437.5 | 744.0 | 73.5 | 122.5 | 2687.0 |
| | Reward | 17.96 ± 6.8 | 17.92 ± 4.4 | 20.35 ± 2.3 | 17.0 ± 5.0 | 19.02 ± 2.2 | 17.43 ± 5.4 |
| | Gap | N/A | -0.0 | 2.4 | -1.0 | 1.1 | -0.5 |
| qbert | Switching cost | 743750.0 | 7437.5 | 744.0 | 73.5 | 122.5 | 1980.5 |
| | Reward | 17685.25 ± 2486.0 | 17038.5 ± 2710.1 | 13836.0 ± 4032.8 | 2193.75 ± 1629.5 | 3838.0 ± 1716.6 | 13791.0 ± 3372.0 |
| | Gap | N/A | -646.8 | -3849.2 | -15491.5 | -13847.2 | -3894.2 |
| riverraid | Switching cost | 743750.0 | 7437.5 | 744.0 | 73.5 | 122.5 | 2325.5 |
| | Reward | 7808.6 ± 308.6 | 7840.3 ± 351.4 | 7872.9 ± 939.1 | 3391.1 ± 821.2 | 4013.5 ± 938.3 | 7052.3 ± 862.7 |
| | Gap | N/A | 31.7 | 64.3 | -4417.5 | -3795.1 | -756.3 |
| road_runner | Switching cost | 743750.0 | 7437.5 | 744.0 | 73.5 | 122.5 | 2736.6 |
| | Reward | 37480.0 ± 8429.9 | 36118.0 ± 6366.6 | 36722.0 ± 7119.7 | 35055.0 ± 7095.8 | 45706.0 ± 8473.7 | 45940.0 ± 8234.1 |
| | Gap | N/A | -1362.0 | -758.0 | -2425.0 | 8226.0 | 8460.0 |

Table 4: We list the value of the switching cost and reward of different criteria when the environment takes 1.5 million steps and 3 million steps. "Reward" corresponds to the absolute value of the reward, and "Gap" denotes the difference between the reward under a specific criterion and "None." And "Switching Cost" corresponds to the switching cost under a criterion at this time step.

