# OpenReview forum: "Deep Q-Learning with Low Switching Cost"
_ICLR.cc/2021/Conference — Reject_

### Official Review · AnonReviewer1 · 2020-10-20
**Interesting setting. A few elements should be improved.**

**Rating:** 5
**Confidence:** 4

**Review:**

In the RL context, this paper aims at designing a generic solution for reducing the number of policy switches during training (called switching cost) while maintaining the performance. This study is done in the context of deep reinforcement learning. A few generic baselines solutions are provided as well as a more complex solution that empirically outperforms the baselines.

Motivation of the paper:
This paper studies an interesting question that is rarely studied in practice. The paper mentions a few applications domains but what would be a very concrete example where the switching cost is important? The paper would benefit from mentioning in what kind of specific application this can be useful to back up the sentences like "Such a requirement is ubiquitous in many applications, such as medical domains, recommendation systems, education, robotics, dialogue agents, etc,".

The approach
The paper investigates different simple switching criteria based on deep Q-networks that are used as baselines and it also proposes an adaptive approach based on the feature distance between Q-networks. I'm not fully convinced by the theoretical justification from a representation learning perspective. What are exactly the extracted feature denoted as f(x) beyond "a representation function that maps the input to a k-dimension vector" and that it is based on the Q-networks. Are these the features at the last layer of the Q-network? In fact, why are they referred to as representation at all in the case of DQN?

Experiments
It seems that the paper does not mention the number of seeds (runs) that are used for the experiments. Otherwise, the experiments back up the claims.

Other comments:
- line 16 of the algorithm: do you mean "h % H_{target}==0" ?
- The structure of the paper might be improved a bit and there are some typos (state encdoer, hyper-paramter, ...)

---

### Official Review · AnonReviewer2 · 2020-10-26
**Proposes an adaptive heuristic for deciding when to update the simulation policy for deep RL**

**Rating:** 5
**Confidence:** 3

**Review:**

######################################

Summary:
In many real world applications for RL such as medicine, there are limits on the number of policies from which we can simulate data. This paper proposes an approach that adaptively decides when to update the simulation policy, based on the difference between it and the current learned policy. Experiments on a medical treatment environment and Atari show that the approach obtains similar performance to on-policy RL with fewer changes of the simulation policy.

######################################

Pros:
1. The proposed approach is straightforward, adaptive, and achieves results comparable to the classical on-policy setting with fewer policy switches on all environments shown. It is also applicable to both model-based and model-free RL.
2. The paper is organized well, and the algorithms are clearly explained.

Cons:
1. I did not find the theoretical justification for the proposed approach to be very convincing for RL, since it is based on a construction in a simplified linear regression case. However, I think this is ok since the paper is application-focused.
2. Based on the results on GYMIC, the proposed approach seems to have much greater variance than the other algorithms, especially at the early stages of training. This is often detrimental for the applications considered, such as medicine, in which robustness is also desirable.

######################################

Overall:
I would lean toward accepting this paper. I am not completely familiar with the literature on RL with low switching cost, but the proposed approach appears to be novel. The experiments show that when combined with Rainbow DQN,  it effectively reduces the switching cost on a range of environments and is based on the training path, requiring less environment-specific hand-tuning than fixed or adaptive interval switching.

######################################

Further comments and questions:
1. How were the six Atari games chosen? How do the different approaches compare in other games?
2. There are a number of typos, e.g. "deno" in the first paragraph of section 3.1.

######################################

Update after reading other reviews and author response:

I have decided to lower my score from 6 to 5, as I agree with Reviewer 3 that more experimental analysis of the method is needed (ablations, sensitivities, etc.) given that the theoretical backing is not convincing. The authors also did not directly answer our questions.

---

### Official Review · AnonReviewer3 · 2020-11-03
**More analysis is required**

**Rating:** 5
**Confidence:** 3

**Review:**

Hi,

First I want to thank authors for putting this manuscript together, and looking into an interesting issue.

*Summary* : Authors proposed an algorithm for switching policies while using Deep Q-learning. The method is based on feature similarity (calculated by cosine similarity) between online and the deployed policy. In addition, they performed experiments in Atari and sepsis simulator.

*Strength* :
1. I believe the paper is well written and easy to follow.
2. An important problem to tackle (well motivated)

*Weakness* :
1. Content : I believe the paper is a good first step, but for a publication analysis should go further, to expand the understanding of the method. For example, I'd like to see the effect of "a" for switching on the performance. It shows how stable/unstable the method will be. The effect of batch size "|B|" on how often we switch, and if we were to use smaller batches (higher variance in sim(B) score) are we going to see a large hit in the performance?
In addition, it is important to check (or discuss) other possible similarity metrics. For example, what if we look at inf norm of the two representation difference, or other similar metrics.

2. Performance : It seems to me that FIX_10^3 has always lower switching cost, and also learns a good policy. I appreciate authors trying 3 different fix number, as mentioned above I will be curious to see effect of "a" as well.

Questions : I think that maybe another good criteria is to measure the distance between the deployed and the online policy, rather than feature representation. It may be the case that features distance determines the policy distance but that's not necessarily true. I was wondering what authors think about this?  (Of course it's hard to measure the distance between the two policy, but maybe using DRL we can measure the distribution distance of the two in a given batch?)


At the end, I would like to say that, the paper is a good a step, but for publication the analysis/ experiments should be more thorough and possibly give insights about how to "formalize" the problem. (which I believe should be the main focus, developing theoretical grounding for low-switching cost problems).

Hope it was helpful,
Thanks.

---

### Official Review · AnonReviewer5 · 2020-11-05
**An important problem but insufficient results**

**Rating:** 4
**Confidence:** 5

**Review:**

This paper studies RL with low switching cost under the deep RL setting. It points out several naive algorithms like switching after a certain number of steps and then propose a new heuristic. This heuristic learns a new policy offline using the experience replay the behavior collected and switches the behavior policy once the similarity of the feature embeddings of the current state by these two policies becomes large. The paper also makes an attempt to provide a theoretical justification for a better understanding of the heuristic. This method might outperform the naive algorithms by some margin, if any. It would be a more interesting manuscript if some stronger results could be provided from the perspective of any of theory, experiments, or applications.

---

### Author Response · Authors · 2020-11-16
**Thank all reviewers for your reviews and suggestions.**

Thank all reviewers for your reviews and suggestions.

In this paper, we take the first step towards designing a generic solution for reducing the switching cost while maintaining performance, which is an important problem when applying the RL algorithms to many real-world scenarios.
We sincerely appreciate all the suggestions and will continue working in this direction in future research.

Although our method may be a little simple, there was no previous study on this important problem, and we empirically and theoretically verify the feasibility of doing so. We believe that there exists great potential in this direction, and further study would promote the application of RL algorithms.

Another thing worth noting is that in real-world applications such as the medical domain, we usually need a deterministic policy, and we also derive a deterministic Rainbow DQN. Although in deterministic Rainbow classic $\epsilon$-greedy exploration is no longer feasible and it is not easy for the policy to explore in the environment, our proposed deterministic Rainbow also achieves empirical success in many scenarios, we believe this also serves as a contribution of our paper.

---

### Decision · Program_Chairs · 2021-01-07
**Final Decision**

**Decision:**

Reject

**Comment:**

This paper studies RL with low switching cost under the deep RL setting. It provides new heuristics for doing so. The reviewers are worrying about whether the problem is important in practice, whether the policies obtained can be used in practice, and the theories might not be strong enough. The paper can be strengthened if better theory and more experiments are provided.